# Hematopoietic Stem Cell Aging: Mechanisms, Microenvironment Influences, and Rejuvenation Strategies

**DOI:** 10.3390/bioengineering12111166

**Published:** 2025-10-27

**Authors:** Jiaqi Cui, Xincan Li, Bin Liu, Cheng Dong, Yun Chang

**Affiliations:** 1Department of Biomedical Engineering, The Hong Kong Polytechnic University, Kowloon, Hong Kong 999077, China; jiaqi.cui@polyu.edu.hk (J.C.); xincan828.li@connect.polyu.hk (X.L.); bin-bme.liu@polyu.edu.hk (B.L.); cheng.dong@polyu.edu.hk (C.D.); 2Shenzhen Research Institute, The Hong Kong Polytechnic University, Shenzhen 518052, China; 3Joint Research Center of Biosensing and Precision Theranostics, The Hong Kong Polytechnic University, Kowloon, Hong Kong 999077, China

**Keywords:** hematopoietic stem cell aging, molecular and cellular mechanisms, microenvironment niche, pluripotent stem cell, rejuvenation strategies

## Abstract

Hematopoietic stem cells (HSCs) are essential for lifelong blood production and immune homeostasis. However, aging induces functional declines in HSCs, leading to hematological disorders, immune dysfunction, and increased susceptibility to malignancies. This review explores the biological underpinnings of HSC aging, highlighting the intrinsic and extrinsic factors that drive this process. We discuss the molecular and cellular mechanisms contributing to HSC aging, including genetic instability, epigenetic alterations, metabolic shifts, and inflammation signaling. Additionally, we examine the role of the bone marrow microenvironment in modulating HSC aging, emphasizing the impact of niche interactions, stromal cell dysfunction, and extracellular matrix remodeling. To advance our understanding of HSC aging, pluripotent stem cell differentiation platforms provide a valuable tool for modeling aged HSC phenotypes and identifying potential therapeutic targets. We review current strategies for HSC rejuvenation, including metabolic reprogramming, epigenetic modifications, pharmacological interventions, and niche-targeted approaches, aiming to restore HSC function and improve regenerative potential. Finally, we present emerging perspectives on the clinical implications of HSC aging, discussing potential translational strategies for combating age-associated hematopoietic decline. By integrating insights from stem cell biology, aging research, and regenerative medicine, this review provides a comprehensive overview of HSC aging and its therapeutic potential. Addressing these challenges will be critical for developing interventions that promote hematopoietic health and improve outcomes in aging populations.

## 1. Introduction

HSCs are the foundation of the hematopoietic system, endowed with the unique abilities of self-renewal and multipotent differentiation to maintain blood cell homeostasis throughout life [1]. The blood system consists of at least ten different types of cells, each with a specific lifespan that needs to be continuously replenished, and HSCs serve as the cornerstone by differentiating into all blood cells, including red blood cells, white blood cells, and platelets, thus ensuring a dynamic balance. In addition, HSCs are engaged in repairing and regenerating the blood system to restore normal hematopoietic function after injury or disease, exemplified by their use in bone marrow transplants to rebuild the patient’s blood system. The HSC niche exists in different tissues throughout development, starting from the aorta-gonad-mesonephros (AGM) region and the yolk sac, and then migrating to the placenta, fetal liver, spleen, and bone marrow. After birth, the bone marrow becomes the predominant site where HSCs are maintained and produce blood, located around blood vessels formed by mesenchymal stromal cells and endothelial cells, often near the trabeculae of the bone. However, with aging, HSCs undergo significant functional decline characterized by reduced regenerative capacity, myeloid-biased differentiation, and increased susceptibility to clonal hematopoiesis and hematological diseases [2]. This aging process is not confined to HSCs themselves but also involves dynamic changes in the bone marrow microenvironment [3] (as illustrated in Figure 1), triggering cascading reactions such as inflammation and metabolic dysregulation, which in turn promote the occurrence of aging-related hematological disorders and malignancies [4]. Understanding the intrinsic mechanisms of HSC aging and extrinsic factors is the current research focus [5,6,7,8]. This not only helps elucidate the essence of hematopoietic system aging but also provides the scientific foundation for developing rejuvenation strategies, ultimately aiming to improve the quality of life for the elderly.

This review aims to integrate the current knowledge framework of HSC aging, systematically reviewing its molecular mechanisms, microenvironmental influences, and potential rejuvenation strategies. We will delve into the interplay of intrinsic and extrinsic factors and critically evaluate the clinical translation potential of rejuvenation strategies, ultimately providing theoretical underpinnings for the prevention and treatment of age-related hematological disorders.

It is important to note that many of the mechanistic insights discussed in this review are derived from murine models. Mouse and human HSCs share high genetic homology, making murine models invaluable for simulating and understanding fundamental physiological processes relevant to human hematopoiesis. However, critical differences exist, particularly in the pace and manifestation of the aging process between species, as mice have a compressed lifespan and accelerated aging trajectory compared to humans. To prevent confusion and clearly distinguish the experimental basis of the findings, we have provided a summary table (Table 1) classifying the species origin (human, mouse, or both) of the key studies cited in this review.

## 2. Functional Deterioration and Systemic Consequences of Aged HSCs

### 2.1. Self-Renewal Depletion

There is a significant decrease in the transplantation reconstruction capacity of aging HSCs compared to the younger population. This functional decline arises from the synergistic effects of cell-intrinsic metabolic defects and a deteriorating mechanical microenvironment. Specifically, a cell-autonomous metabolic disorder involving Socs3-mediated insulin resistance delays the initiation of glycolysis by impairing the membrane localization of the GLUT1 transporter. This, in turn, prevents HSCs from mounting a timely response to regenerative demands [66]. This process is exacerbated by microenvironmental changes in mechanical properties: with age, bone marrow stromal stiffness increases significantly from 3 to 8 kPa in young adulthood, inhibiting mesenchymal stromal cell (MSC) CXCL12 secretion through activation of the mechanosensitive proteins Yap/Taz, and thus disrupting the maintenance of the HSC resting state [9]. This “ecological sclerosis” creates a positive feedback loop—metabolic defects reduce HSC adaptability, and mechanical signaling abnormalities further limit functional recovery. The immediate pathological output is age-related anemia. Self-renewal depletion is often accompanied by clonal hematopoiesis, a hallmark of human hematopoietic aging and is not typically observed in mouse models. Under self-renewal depletion, some clones of HSCs that have gained a proliferative advantage escape aging stress through epigenetic reprogramming and dominate the pool of depleted HSCs [10,11]. These clones not only have the advantage of autonomous proliferation, but also activate osteoclast differentiation by secreting factors such as RANKL, while inhibiting the osteogenic potential of MSCs, forming the HSC–bone metabolism imbalance axis, thus actively driving the progression of osteoporosis [67]. These findings overturn traditional knowledge and reveal that aging HSCs are not a passive result of bone loss but of actively remodeling the bone microenvironment through clonal hematopoiesis as an initiating factor of bone metabolism disorders.

This clonal evolution is not limited to bone metabolism imbalance but is profoundly associated with leukemic transformation. Under the synergistic effect of a chronic inflammatory microenvironment, DNMT3A mutation disrupts the HSC proliferation–differentiation balance by activating the NF-κB pathway through aberrant DNA methylation, while TET2 deletion promotes clonal amplification through enhanced STAT3 phosphorylation [45,46]. Meanwhile, TSP-1 in senescence-associated secretory phenotype (SASP) inhibits apoptosis and maintains the stem cell properties of mutant clones by activating TGF-β signaling [61].

However, an interesting paradox has been shown to us in recent years by a study in which clinical observations showed that TET2-deficient clones were associated with milder anemia in patients with low-risk myelodysplastic syndromes [12]. This suggests that specific mutations may act as ‘adaptive compromises’ for aging HSCs in response to the stresses of aging—trading off long-term leukemia risk for short-term hematopoietic maintenance. This evolutionary trade-off highlights the multi-stage nature of leukemic transformation: early mutations alleviate hematopoietic failure, while persistent inflammatory stimuli ultimately drive clonal evolution to a fully malignant phenotype.

### 2.2. Myeloid Differentiation Bias

Myeloid differentiation skew and attenuation of lymphoid lineage potential in aging HSCs are central drivers of immune aging. Aging HSC differentiation shows a significant myeloid skew: a decline in lymphoid lineage potential (especially B-lymphocytes) and an increase in granulocyte or monocyte output. Single-cell transplantation revealed a reduced proportion of gonadotroph-biased stem cells and an expansion of myeloid-restricted progenitor cells in the aged HSCs [9]. In terms of molecular mechanisms, abnormally elevated HDAC3 activity leads to deacetylation at the H4K77 site, which directly inhibits the expression of key genes for lymphoid lineage differentiation such as EBF1 and PAX5 [68]. This epigenetic error is exacerbated by metabolic heterogeneity: approximately 30% of HSCs (CD150low) maintain functional homeostasis through autophagy, whereas 70% of HSCs (CD150high) are forced to differentiate toward the myeloid spectrum due to mitochondrial autophagy defects that lead to the accumulation of reactive oxygen species (ROS) [13,66]. Clinically, the associated pathological consequences are significant: first, lymphoid output failure leads to a significant reduction in TCR/BCR diversity, which in turn leads to a 4-fold decrease in influenza vaccine antibody titers [68]. Simultaneous dysfunction of the myeloid system, loss of tumor immune surveillance, efficiency of tumor neoantigen recognition, and a 40% increase in melanoma metastases [61].

Notably, the mechanisms driving this skewed differentiation remain controversial. Single-cell epigenomic data support HSC autonomous program disorders, such as increased H3K27me3 deposition in the promoter region of the gonadotropin gene [12]. However, the potential influence of the extrinsic microenvironment is supported by other transplantation studies, which show that aged systemic or local niche factors can impair the function of young HSCs and promote myeloid skewing, although the effect may be context-dependent and not always override strong cell-intrinsic programming [14]. This highlights that the myeloid bias in aging is a complex interplay of cell-autonomous changes and extrinsic pressures [47]. Based on this, there seems to be a tendency for metabolic adaptation in this myeloid differentiation tendency: aged HSCs maintain energy homeostasis by activating fatty acid oxidation (PPARγ pathway), and this metabolic switch itself naturally promotes myeloid differentiation through epigenetic reprogramming, which tightly combines HSC’s own regulatory mechanisms with external microenvironmental signals to drive the process of myeloid differentiation.

### 2.3. Dysfunctional Differentiation and Platelet Hyperactivity

Dysregulation of cell polarity in aging HSCs leads to impaired clearance of damaged mitochondria1, and dysregulation of the RNA-binding protein FUS (FUsed in Sarcoma), particularly its aberrant liquid–liquid phase separation (a physical process underlying biomolecular condensate formation), further drives aged HSCs to skip the progenitor cell stage and differentiate directly into megakaryocytic progenitor cells (MkP) [48] (as illustrated in Figure 2). This age-related disruption in FUS phase behavior alters chromatin topology and compromises the fidelity of gene regulation, thereby promoting aberrant differentiation pathways. This aberrant differentiation pathway produces highly reactive platelets: upregulation of P-selectin and integrin αIIbβ3 expression on their surface increases the risk of venous thrombosis by 300% in older individuals [69]. More importantly, these platelets are not only coagulation mediators, but also amplifiers of the inflammatory cascade-the miR-126 they carry forms a “platelet-endothelial-inflammatory axis” through activation of the NF-κB pathway in vascular endothelial cells, a mechanism which explains the potential value of antiplatelet therapy in ameliorating chronic inflammation in the elderly.

## 3. Cell-Autonomous Mechanisms Driving HSC Aging

### 3.1. Genomic Instability

HSCs are challenged by a significant decrease in genomic stability during aging. The conventional view is that the resting state of HSCs is a protective mechanism for their long-term maintenance of genomic integrity. However, a study by Beerman et al. overturns this perception, revealing that the resting state is instead a key factor in the increased genomic instability of aging HSCs [10]. They found a significant accumulation of DNA strand breaks in aging HSCs of mice due to extensive attenuation of DNA damage repair and response pathways (e.g., non-homologous end-joining and down-regulation of homologous recombination-associated gene expression) in resting HSCs. When resting HSCs enter the cell cycle, these pathways are activated and repair damage. This mechanism of “resting-dependent repair inhibition” provides a rational explanation for the accumulation of premalignant mutations in aging HSCs. It is worth noting that this attenuation of repair capacity is dynamically reversible: when HSCs enter the cell cycle, DNA repair mechanisms are reactivated and breaks are repaired, suggesting that quiescence is both a key strategy for HSCs to maintain long-term survival and a source of aging HSC-associated DNA damage accumulation. Nevertheless, Flach et al. [11] revealed another layer of complexity—aged HSCs face increased replication stress even when they enter the proliferative state. This stress stems from the destabilization of the DNA replication fork due to reduced expression of the MCM helicase component, which in turn triggers chromosome breakage. Notably, replication stress is amplified under stressful conditions such as transplantation, resulting in irreversible attrition of aged HSCs. Together, these two studies suggest that the quiescent state of HSCs is protective in youth but becomes a ‘double-edged sword’ of genomic damage in old age.

This mechanism of damage accumulation was further extended in the study of He et al. [49]. They found that DNA damage promotes the nuclear translocation of the MCPH1 protein through KAT7-mediated acetylation, leading to the deregulation of the inhibitory effect of MCPH1 on the necroptotic apoptotic pathway (RIPK3-MLKL) in the cytoplasm. This finding directly links DNA damage to programmed cell death mechanisms and explains why functional decline occurs in aged HSCs in parallel with accumulated genomic damage. Interestingly, the metabolic–epigenetic coupling hypothesis proposed by Hajishengallis et al. [15] provides a new perspective in this regard: dysregulation of the energy metabolism required for DNA repair may exacerbate the decline in repair efficiency, creating a vicious cycle.

Yet, existing studies still lack a fine-grained picture of the spatial and temporal dynamics of DNA damage accumulation, and most of them are still based on population-level analyses, which may mask the heterogeneity among HSC subpopulations. The recent findings of Totani et al. [16] provide clues in this regard: a subpopulation of HSCs with a high mitochondrial mass (labeled as GPR183+) still maintains strong regenerative capacity in the aged bone marrow, which suggests that there may be subpopulation specificity in the accumulation of genomic damage.

### 3.2. Epigenetic and Metabolic Disorders

Disturbances in the epigenetic regulatory network are another central feature of aging HSCs. He et al. [17] found that the aberrant accumulation of the TRMT6-TRMT61A complex in aging HSCs does not occur through the classical tRNA m1A methylation pathway, but rather through the generation of the 3’-tiRNA-Leu-CAG-activated RIPK1-RIPK3-MLKL necroptotic apoptosis pathway. This finding overturns conventional knowledge and suggests that RNA-modifying enzymes can regulate cell fate through non-classical pathways. Correspondingly, Gong et al. [68] used a novel Iseq-Kac technique to reveal that a significant reduction in histone H4K77 acetylation (H4K77ac) was associated with elevated HDAC3 activity in aged HSCs, an epigenetic alteration that directly contributes to the reduced ability of lymphoid lineage differentiation. Notably, the reduction in H4K77ac is not a global phenomenon but specifically affects the degree of chromatin opening of genes related to B-cell differentiation (e.g., EBF1). This observation implies a potential hierarchical organization in the regulatory drivers of epigenetic remodeling.

The heterogeneity of epigenetic regulation is further exemplified in the study of MCPH1 [49]. the classical function of MCPH1 to maintain genome stability in the nucleus forms a spatially segregated regulatory network with its novel function to inhibit necrotic apoptosis in the cytoplasm. This regionalized dysfunction suggests that epigenetic regulators may achieve multilevel control of HSC fate through dynamic changes in subcellular localization. Song et al. [18], on the other hand, revealed a novel mechanism of metabolic-epigenetic coupling: aberrant CD38-mediated NAD+ metabolism leads to a reduction in SIRT1 deacetylase activity in aging HSCs through dysregulated mitochondrial calcium signaling, which in turn affects epigenetic modifications of key transcription factors such as FoxO.

### 3.3. Inflammations

HSCs suffer irreversible damage to their self-renewal capacity when exposed to acute inflammatory stimuli, and this functional depletion phenomenon exhibits a sustained amplification effect in the context of chronic inflammation. Bogeska et al. [19] found by tracking a mouse model that transient inflammatory exposures can lead to a permanent reduction in functional HSCs, and that even one year after inflammation has subsided, the regeneration capacity of the resting-state HSCs is still not restored. The molecular basis of this irreversible damage stems from a systemic breakdown of the epigenetic regulatory network: acute inflammation induces aberrant activation of the ERK-ETS1 pathway via TNF-α-induced IL27Ra signaling, which directly inhibits the self-renewal capacity of HSCs and promotes myeloid differentiation bias [48]. In contrast, long-term chronic inflammation triggers even more profound epigenetic reprogramming—aberrant phase separation of FUS proteins leads to functional fusion of chromatin topology domains (TADs), disrupting the precise regulation of stem cell maintenance genes such as HOXA9 [50]. Cui’s team further revealed that aging HSCs undergo a unique 3D genomic remodeling under LPS stimulation, which is manifested by the enhancement of abnormal binding of CTCF insulators in the promoter region of myeloid genes, and that this epigenetic ‘imprinting’ causes HSCs to continue to exhibit myeloid differentiation in the absence of stimulation [20], creating a pathologic state similar to an epigenetic memory.

Remarkably, functional depletion of HSCs not only stems from cell-autonomous injury but is also synergistically influenced by the bone marrow microenvironment. Young et al. found that heterogeneity of Kitl/Igf1 expression in MSCs independently predicted the lymphoid potential of HSCs [21], suggesting that even in the inflammation-induced overall functional decline, the local microenvironment can still regulate the direction of HSC differentiation through specific signaling pathways (e.g., SCF/c-Kit). Differences in the microenvironment of different regions of the bone marrow may be an important reason for the varying degrees of hematopoietic decline in elderly patients. This model suggests that when pro-inflammatory signals (e.g., TNF-α) and pro-lymphoid differentiation signals (e.g., Kitl) counteract each other locally, HSCs may enter a state of functional standstill—losing self-renewal capacity while exhibiting a bias toward myeloid cell production. Collectively, these interactions can create a vicious cycle: inflammation leads to epigenetic abnormalities in stem cells, which impair their function; these dysfunctional stem cells in turn may release more pro-inflammatory substances, which further destabilize the bone marrow environment, ultimately contributing to the functional decline of the entire hematopoietic system.

## 4. Microenvironment Niches of HSC Aging

### 4.1. Niche Cell-Mediated Inflammatory Activation in Aging

Within the aged bone marrow, elevated concentrations of pro-inflammatory cytokines establish an inflamed niche microenvironment. This inflammatory milieu is further remodeled by niche cells expressing inflammation-associated receptors, ultimately contributing to HSC aging. As a major niche cell type, endothelial cells (ECs) express high levels of TLR4 and Myd88. During systemic challenges such as LPS exposure or infection, aged ECs function as major producers of G-CSF, thereby driving emergency granulopoiesis [22]. Additionally, they promote vascular leakiness and angiogenesis, mediated by increased sinusoidal density and integrin αVβ3 expression [23]. Critically, experimental evidence demonstrates that aged ECs directly induce myeloid differentiation bias and impair HSC self-renewal [24]. This was established both by co-culturing young HSCs with aged ECs and by infusing aged endothelium into young recipients. Furthermore, stromal cells within the aged niche exhibit a significant elevation in inflammatory output, characterized by increased expression of chemokines Cxcl2 and Cxcl5, complement factors Cfd, Cfb, C4b and C3, and key pro-inflammatory cytokines IL-1β, IL-6 [25]. Studies indicate that in aged bone marrow, plasma cells activate stromal cells through TLR-driven secretion of inflammatory factors IL-1 and TNF-α. These activated stromal cells then express myeloid-promoting factors CSF1 and IL-1β, establishing an inflammatory network that enhances myeloid differentiation of HSCs while suppressing lymphoid lineage development [70] (see Figure 3). Collectively, these findings establish the aged niche as a pro-inflammatory hub. Plasma cells trigger this cascade by activating stromal cells via TLR-driven IL-1and TNF-α secretion. Critically, aged ECs characterized by heightened TLR4 expression act as potent amplifiers, which drive excessive G-CSF production, emergency granulopoiesis, vascular leakiness, and angiogenesis. Simultaneously, activated stromal cells intensify the inflammatory milieu through chemokine, complement and cytokine release. This TLR4-driven synergy between endothelial cells and stromal cells creates a potent inflammatory network that impairs HSC self-renewal, enforces myeloid-biased differentiation, and ultimately propagates the functional decline of the hematopoietic system in aging.

### 4.2. Extracellular Matrix Dysregulation in Aging

Beyond inflammation, extracellular matrix (ECM) represents another fundamental alteration within the aging bone marrow niche that directly contributes to HSC dysfunction. During niche aging, progressive accumulation and cross-linking of matrix proteins elevate tissue rigidity, thereby mechanically activating the transcriptional co-regulators YAP/TAZ in MSCs [9,62]. Proteoglycan, another major component of the ECM, would be altered in aged niches, disrupting the gradient maintenance of key growth factors like IL-3 and GM-CSF and impairing HSC anchorage and homeostasis [51] (see Figure 3). Additionally, imbalanced matrix metalloproteinases (MMPs) and their inhibitors (TIMPs) generate pro-inflammatory matrix fragments, interfering with HSC–stromal interactions [26].

### 4.3. Metabolic Reprogramming and Additional Niche Alterations

Aged niches exhibit significant marrow adipose expansion [27]. On the one hand, these adipocytes alter the metabolic state of the microenvironment, directly impacting HSC regulation through altered secretion patterns of factors like leptin and adiponectin [28]. Furthermore, the accumulation of adipocytes physically encroaches upon HSC niche space and may disrupt the interactions between HSCs and other niche cells (see Figure 3). Another dimension of change involves degeneration of the sympathetic nervous system (SNS) innervation in the bone marrow [27]. The loss of SNS nerve fibers or diminished β3-adrenergic receptor signaling was shown to directly drive HSC aging. Simultaneously, a third critical change is the reduction in insulin-like growth factor-1 (IGF-1) within the middle-aged bone marrow microenvironment, which has been identified as a key factor triggering HSC senescence [71]. Activation of the IGF-1 signaling pathway is essential for maintaining HSC proliferation, differentiation, and function; its decline consequently leads to diminished HSC functionality.

## 5. Induced Pluripotent Stem Cell (iPSC) Differentiation Platform for HSC Aging Investigation

Understanding the complex biology of HSC aging requires models that can replicate key cellular and molecular changes. Induced pluripotent stem cell (iPSC)-based differentiation platforms offer a controllable and scalable in vitro system to study age-related HSC phenotypes, dissect regulatory mechanisms, and explore potential therapeutic strategies. This part highlights recent advances using iPSC models to investigate HSC aging and develop targeted interventions.

### 5.1. Modeling HSC Aging Using iPSC-Based Differentiation Platforms

iPSC-based differentiation platforms have emerged as powerful tools for modeling HSC aging in vitro, providing insights into key phenotypic features and regulatory mechanisms [8]. Through directed differentiation of induced pluripotent stem cells (iPSCs), researchers have been able to recapitulate age-related characteristics of HSCs, including diminished self-renewal, myeloid-biased differentiation, and impaired homing capacity [52]. For instance, 3D embryoid body culture systems combined with Wnt signaling modulation have successfully mimicked the endothelial-to-hematopoietic transition (EHT), generating HSC-like cells that exhibit aging-associated functional deficits [29,53]. These models also highlight the synergistic effects between intrinsic factors and extrinsic influences. Xenotransplantation studies further underscore the role of the microenvironment, demonstrating that aged niches can accelerate the decline of young HSCs, whereas young HSCs can partially rejuvenate aged hematopoietic systems [13,14,30]. These findings inform the design of biomimetic in vitro systems that incorporate niche-derived regulatory signals. Advances in single-cell sequencing and machine learning are improving the resolution with which these models can capture the heterogeneity of HSC aging [72]. However, recreating truly long-term, multilineage reconstituting aged HSCs in vitro remains a major challenge, necessitating further refinement of cell–niche signaling simulations within differentiation protocols.

### 5.2. Decoding HSC Aging Through Single-Cell and Multi-Omics Technologies

The integration of single-cell transcriptomics with pluripotent stem cell-based differentiation platforms has significantly advanced our understanding of the regulatory networks underlying HSC aging. By combining single-cell RNA sequencing with machine learning approaches, researchers have identified key hallmarks of aged HSCs, including dysregulated metabolic pathways, altered division patterns marked by reduced self-renewal and myeloid bias, and dynamic changes in gene expression profiles, such as upregulated inflammatory signaling and impaired DNA repair capacity [31,54,55]. Single-cell studies show that aging reshapes the bone marrow niche, with stromal cells producing more pro-inflammatory factors and diverse Clusterin-positive HSCs accumulating [32]. Furthermore, integrative single-cell multi-omics has uncovered age-related alterations in chromatin accessibility and transcription factor activity [73]. When combined with lineage-tracing tools, these technologies enable the mapping of transcriptional states to clonal fate at single-cell resolution [33]. Together, these high-resolution approaches facilitate the construction of aging-specific molecular atlases and guide functional validation via pluripotent stem cell differentiation systems, such as CRISPR-based screening of candidate regulators [34], thereby accelerating the development of targeted interventions for HSC aging.

### 5.3. Pluripotent Stem Cell-Based Platforms for Anti-Aging Drug Discovery and Functional Restoration of HSCs

Using HSC models derived from iPSCs, researchers have developed high-throughput drug screening platforms that have successfully identified multiple small-molecule compounds capable of reversing aging-associated HSC phenotypes [35,56,74]. These include candidate drugs targeting inflammatory signaling pathways, such as NF-κB, and metabolic regulators involved in processes like glutamine metabolism. By recapitulating the EHT through 3D organoid cultures combined with chemical modulation, partial restoration of HSC function has been achieved—most notably in correcting aging-related myeloid bias and improving homing capacity [53,57]. Furthermore, the integration of gene editing technologies such as CRISPR with stem cell platforms has enabled precise manipulation of age-associated epigenetic markers, offering targeted strategies for rejuvenating HSC functionality [75].

## 6. Rejuvenation Strategies of HSC Aging

As our understanding of HSC aging deepens, growing attention has turned toward strategies to restore HSC function and delay hematopoietic decline. Recent advances highlight diverse approaches targeting both intrinsic cellular mechanisms and the aged microenvironment. This part outlines emerging therapeutic strategies, including molecular, metabolic, inflammatory, and transplantation-based interventions aimed at rejuvenating aged HSCs and improving hematopoietic health during aging.

### 6.1. Targeting Intrinsic Aging Mechanisms to Restore HSC Function

Recent studies have identified multiple strategies to rejuvenate aged HSCs by targeting intrinsic molecular pathways. Epigenetic modulation has shown promise, particularly by restoring the dynamic balance of histone modifications such as H3K4me3 and H3K27me3, thereby reversing age-associated transcriptional silencing [36,76]. Mitochondrial function and redox homeostasis can also be improved through activation of antioxidant selenoproteins, which help mitigate oxidative stress [37]. Proteostasis restoration achieved by enhancing proteasome activity reduces the accumulation of misfolded proteins, alleviating stress in aged HSCs [63]. Interventions targeting cell cycle and DNA repair mechanisms, such as the clearance of age-related extrachromosomal circular DNA or the activation of DNA damage response pathways, have also demonstrated rejuvenating effects [58]. Additionally, reprogramming strategies using defined transcription factors can restore youthful transcriptional programs. Targeting intrinsic signaling pathways, including Yap/Taz and GATA2, has further been shown to enhance self-renewal capacity and delay aging phenotypes in HSCs [35,38]. Furthermore, interventions aimed at enhancing self-renewal can directly contribute to expanding the HSC pool. For instance, transient activation of Yap signaling in mouse models has been shown to promote the ex vivo expansion of HSCs without exhausting them [77]. Similarly, reprogramming strategies using defined factors hold potential for expanding the population of human hematopoietic progenitor cells with long-term reconstitution capacity in vitro [59]. These approaches highlight that targeting core self-renewal pathways offers a viable route to concurrently improve both the quantity and quality of HSCs. Collectively, interventions targeting epigenetic regulation, redox homeostasis, and proteostasis offer promising avenues to reverse the cell-autonomous drivers of HSC aging. A comprehensive summary of these intrinsic aging hallmarks and their targeted interventions is visually synthesized in Figure 4 (adapted from [4]).

### 6.2. Engineering and Modulating the Aged Niche to Rejuvenate HSC Function

Accumulating evidence suggests that the aging bone marrow microenvironment contributes significantly to HSC dysfunction by altering intercellular interactions and activating signaling pathways [9,64]. Bioengineered niches have emerged as promising tools to reverse these effects by mimicking the biophysical and biochemical characteristics of the native microenvironment [9]. Strategies such as tuning matrix stiffness, constructing artificial extracellular matrices, and utilizing synthetic scaffolds have been shown to activate regenerative pathways, thereby restoring HSC self-renewal capacity [39]. Targeting niche-specific defects, such as the FGF23-mediated bone-anemia axis or reprogramming inflammatory signals, can also correct myeloid-biased differentiation and enhance transplantation efficiency [40,78]. In addition, systemic factors from young blood or pharmacological agents have been shown to reshape the molecular landscape of the aged niche, restoring HSC homeostasis [35]. Importantly, preserving key niche signals, such as CXCL12 secretion, remains critical for maintaining the regenerative potential of aged HSCs. In vivo, strategies like co-transplantation with MSCs not only improve homing and engraftment but also leverage the supportive paracrine signals from MSCs to facilitate the expansion of transplanted HSCs within the host [41]. These multifaceted approaches that modulate cellular components, mechanical cues, and secreted factors within the niche offer a broad therapeutic framework for reversing HSC aging.

### 6.3. Therapeutic Strategies Targeting Inflammation and Metabolic Pathways to Rejuvenate Aged HSCs

Aging-associated inflammation plays a pivotal role in HSC dysfunction. He et al. demonstrated that the TNF-α → ERK → ETS1 → IL-27Ra signaling axis mediates pro-inflammatory responses that contribute to HSC aging by reducing self-renewal capacity and promoting a bias toward myeloid differentiation [48]. Inhibition of such inflammatory signaling has been shown to restore self-renewal capacity and mitigate lineage skewing in aged HSCs. Among anti-aging compounds, the natural molecule 2,3,5,4’-tetrahydroxystilbene-2-O-β-D-glucoside (TSG) has demonstrated significant efficacy; oral administration enhances the regenerative potential of aged HSCs and improves hematopoietic function in elderly mice [35]. Additionally, therapeutic strategies targeting the mitochondrial–endoplasmic reticulum metabolic axis have been shown to regulate HSC proliferation and enhance stress resistance, thereby delaying aging phenotypes [18,42]. Senolytic therapies, aimed at selectively eliminating senescent cells, and transplantation of young HSCs have also been shown to rejuvenate immune function by reducing chronic inflammation and introducing pro-regenerative, anti-aging factors into the microenvironment [43,65]. Overall, these interventions highlight the therapeutic potential of targeting inflammation, metabolism, and niche remodeling to restore function in aged HSCs.

### 6.4. HSC Transplantation and Combined Interventions: Emerging Strategies for Reversing Hematopoietic Aging

Recent advances in cellular therapy have demonstrated that transplantation of young HSCs into aged recipients can partially reverse aging phenotypes [13]. A seminal study by Yuan et al. provided a direct mechanistic insight, demonstrating that young donor HSCs can transdifferentiate into functional bone marrow niche cells, thereby actively revitalizing an aged or damaged niche (as illustrated in Figure 5) [60]. However, the rejuvenating effects are often limited by the aged bone marrow niche, which presents a non-permissive microenvironment for transplanted HSCs. Conversely, aged HSCs display only modest functional recovery even when placed in a young niche, suggesting a complex interplay between intrinsic aging mechanisms and extrinsic environmental cues [14]. Beyond direct transplantation, ex vivo expansion of HSCs prior to transplantation represents a crucial strategy to increase the number of functional cells. Obtaining a sufficient quantity of HSCs, especially from aged donors, is a significant challenge in clinical transplantation. Therefore, developing expansion technologies that do not compromise the long-term reconstitution capacity of HSCs is paramount. At the clinical level, the use of the small molecule UM171 to expand human umbilical cord blood HSCs ex vivo has significantly improved patient engraftment outcomes [44]. However, the efficacy of these strategies specifically for expanding aged human HSCs remains to be explored.

Combined intervention strategies have shown promise in overcoming the limitations of reversing hematopoietic aging. For instance, in the context of aging, inhibition of the inflammatory TNF-α/IL-27Ra signaling pathway can alleviate HSC aging phenotypes, while the combination of transplantation with mild endoplasmic reticulum stress inducers or specific anti-aging compounds may enhance HSC self-renewal and stress resistance, thereby promoting functional rejuvenation [79,80]. Targeted removal of senescent HSC subpopulations followed by young HSC transplantation has also been shown to significantly improve hematopoietic reconstitution in aged hosts [13]. Niche-targeted approaches, such as co-transplantation with MSCs, have been effective in restoring HSC function and promoting post-transplant homing and engraftment [81]. Furthermore, dietary restriction and metabolic interventions have been reported to improve the engraftment efficiency of aged HSCs, highlighting nutrient signaling as a potential axis for combinatorial rejuvenation strategies [82]. These findings, including the niche-rebuilding capacity of HSCs shown in Figure 5 [60], support a multi-pronged therapeutic framework integrating HSC transplantation with inflammation, niche, and metabolic modulation to counteract hematopoietic aging.

## 7. Conclusions and Perspective

HSC aging is a multifactorial process driven by intrinsic molecular alterations and extrinsic microenvironmental changes, ultimately leading to impaired regenerative capacity, lineage bias, and increased susceptibility to hematological disorders. Advances in single-cell technologies, PSC-based modeling, and omics profiling have greatly expanded our understanding of the regulatory networks underlying HSC aging. Moreover, innovative therapeutic strategies, ranging from epigenetic and metabolic modulation to niche engineering and combinatorial transplantation, have shown promising potential to reverse age-related dysfunction. Looking forward, integrating high-resolution aging models with precision editing and personalized interventions will be critical for translating these findings into clinically viable anti-aging therapies for the hematopoietic system. Continued interdisciplinary efforts will be essential to overcome current limitations and realize the full potential of HSC rejuvenation in regenerative medicine and aging-related disease prevention.

While murine models have been instrumental in elucidating the cellular and molecular basis of HSC aging, it is crucial to acknowledge that mice exhibit a compressed lifespan and accelerated aging trajectory compared to humans. These differences may influence the manifestation of aging phenotypes, such as clonal hematopoiesis and inflammatory niche remodeling. Future studies leveraging humanized models or longitudinal human data will be essential to validate these mechanisms in the context of human aging.

## Figures and Tables

**Figure 1 bioengineering-12-01166-f001:**
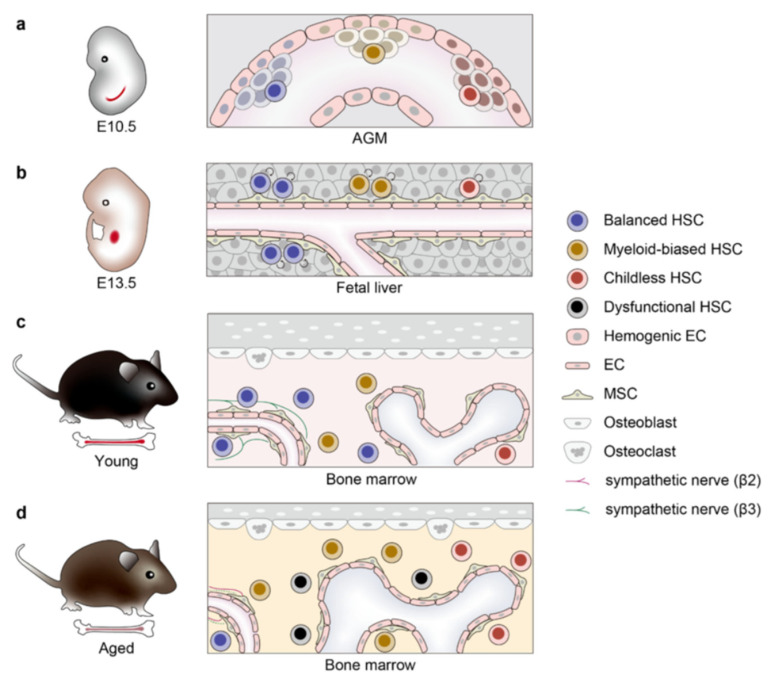
HSC development, aging, and clonal dynamics. Hematopoietic stem cell (HSC) developmental trajectory across murine lifespan. (**a**) E10.5 mouse aorta-gonad-mesonephros (AGM) region: Early HSC emergence with heterogeneous subpopulations, including balanced HSC, myeloid-biased HSC, and childless HSC. (**b**) E13.5 fetal liver stage: HSC migration to the fetal liver, marked by rapid expansion and lineage commitment within a permissive stromal microenvironment. (**c**) Young adult bone marrow niche: Stabilized HSC residency in the hypoxic endosteal region, maintaining quiescence and balanced differentiation under steady-state regulation. (**d**) Aged bone marrow microenvironment: Accumulation of dysfunctional HSCs exhibiting myeloid skewing, reduced regenerative capacity, and senescence-associated phenotypes. Figure adapted from [3].

**Figure 2 bioengineering-12-01166-f002:**
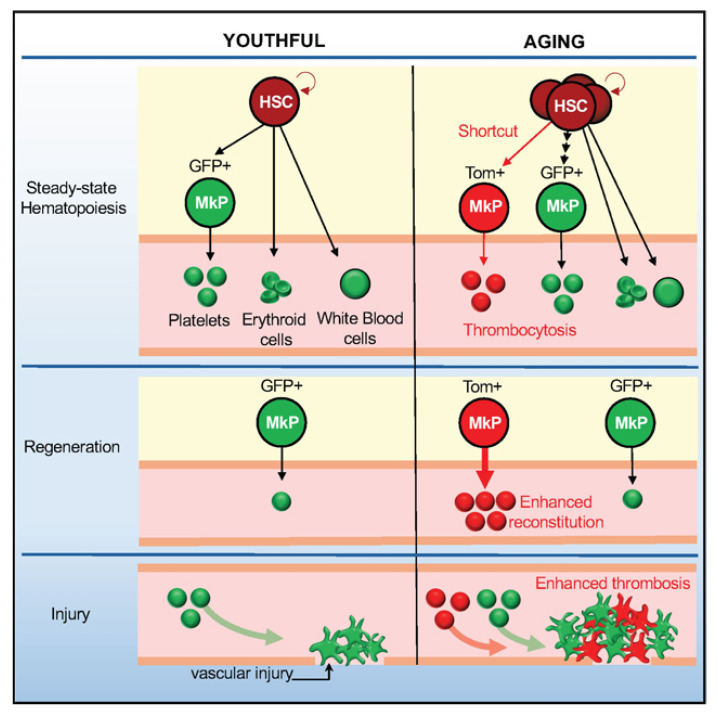
A schematic model comparing megakaryopoiesis in youthful and aging states. In the aging state, HSCs take a shortcut differentiation path to generate Tom+ MkPs, which leads to thrombocytosis, enhanced reconstitution, and exacerbated thrombosis. Figure adapted from [47].

**Figure 3 bioengineering-12-01166-f003:**
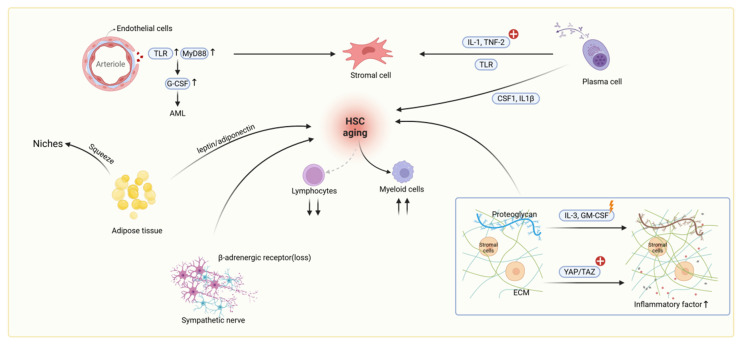
The aged bone marrow microenvironment impairs HSC function through integrated inflammatory, structural, and cellular alterations. This schematic illustrates the key cellular and molecular changes within the bone marrow niche that collectively contribute to HSC aging and functional decline.

**Figure 4 bioengineering-12-01166-f004:**
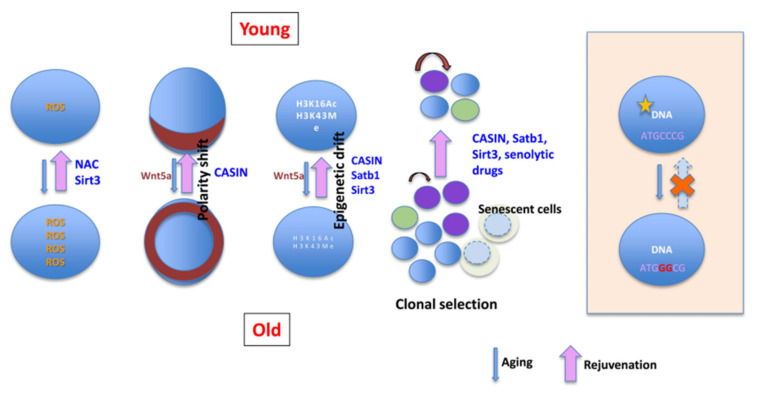
Mechanisms of HSC aging and potential rejuvenation strategies. This schematic illustrates the contrast between young and old HSCs. Key hallmarks of aging include the accumulation of ROS, epigenetic drift, and a loss of cellular polarity. The process of clonal selection during aging can lead to the expansion of dysfunctional, senescent cells. Potential rejuvenation strategies aim to target these senescent cells or reverse age-associated alterations, promoting functional recovery toward a more youthful state. Figure adapted from [4].

**Figure 5 bioengineering-12-01166-f005:**
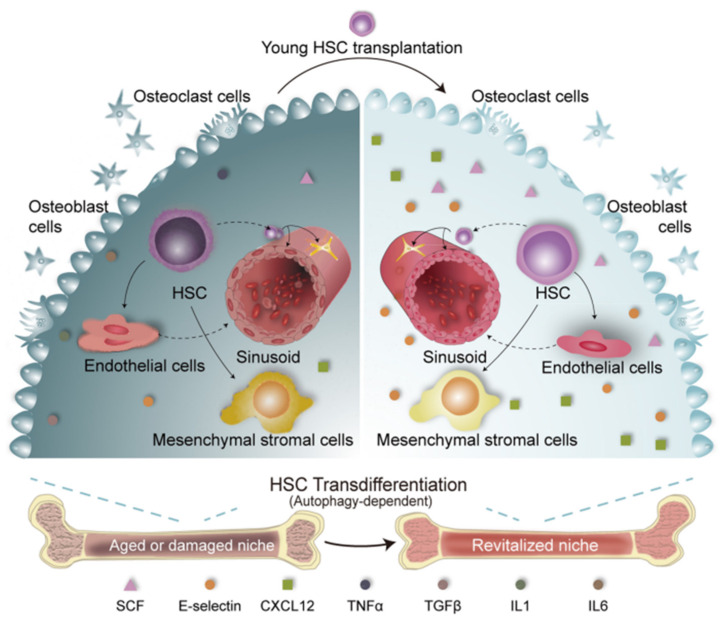
Schematic model of young donor HSC-mediated rejuvenation of the aged bone marrow niche. Young donor HSCs are transplanted into aged or damaged bone marrow. These HSCs not only reconstitute the hematopoietic system but also transdifferentiate into functional niche cells. This transdifferentiation revitalizes the niche by restoring niche factors and reducing inflammation, thereby improving hematopoietic function. Figure adapted from [60].

**Table 1 bioengineering-12-01166-t001:** Species origin of key cited studies.

Experimental Model	Reference Number
Mouse	[5,9,10,11,12,13,14,15,16,17,18,19,20,21,22,23,24,25,26,27,28,29,30,31,32,33,34,35,36,37,38,39,40,41,42,43,44]
Human	[1,2,6,45,46,47,48,49,50,51,52,53,54,55,56,57,58,59,60]
Both	[61,62,63,64,65]

## Data Availability

Not applicable.

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
