# Peer review of "Hematopoietic Stem Cell Aging: Mechanisms, Microenvironment Influences, and Rejuvenation Strategies"

_bioengineering, 2025, doi:10.3390/bioengineering12111166_

Round 1

Reviewer 1 Report

Comments and Suggestions for Authors

Dear Authors,

here are my comments on your review.

  1. Figures 1, 2, 4 and 5 are not adopted (as mentioned in figure captions), but bluntly copied from the cited articles. This is absolutely unacceptable. Some figures even bear some indications, used in the original article, but useless in this review. You can't just copy someones' figures, it's straight plagiarism!!!
  2. I've detected at least 27 inappropriate references cited, which is roughly 1/3 of all references in the text. I do believe this is inappropriate for any rext pretending to be true. Some references describe another cell type, not the one, you're referring to in the text. Another ones cover unrelated scientific field and do not describe HSC or hematopoiesis. There are some citation errors I'd found ([citation number] - line number): [9]-77, [10]-116,[17]-130, [20] -132, [21]-144, [31]-240, [32]-260, [40]-294, [41]-297, [8]-323, [43]-326, 44, 45]-328, [48] -335, [49]-346, [55]-355, [56-58] - 360, [44, 59]-365, [60]-368, [61]-380, [65] -387, [73]-434, [75, 76] - 452, [77]-456.
  3. From the previous comment follows the next one - some of the statements in the review are not supported by appropriate references. Therefore, they rise doubts. For example, part 4.3 does not contain single justified reference!
  4. Figure 3 should demonstrate clonal dynamics, however there no such data on it or in its captions.
  5. Lines 77-81 are unclear.
  6. Lines 130-132: in [20] the consistency of young features of HSC had been demonstrated, on the contrary to what you're saying in the text. Therefore, the followed conclusions are unclear and unproved.
  7. Part 1.3 is named 'Abnormal polarity regulation' while there no such data in it. Therefore, all part 1.3 is a bit senceless.
  8. Line 140: what is FUS phase, where did it come from and how did it related to HSC ageing?
  9. Line 172: there should be MCM helicase, not deconjugase.
  10. Lines 204 and 240: what 'gonadal differentiation' and 'gonadotrophic potential' of HSC did you mean?
  11. Lines 244 and 251 need references.
  12. Part 4. Did you mean iPS or ESC? From the following text I've figured out you pay little attention on correct type of precursors, sometimes you're mixing them up. I think, these two types of stem cells should be clearly described separately. 
  13. Lines 452-456 do not describe ageing or its reversal, as they sould be.

Reviewer 2 Report

Comments and Suggestions for Authors

The review is mostly dedicated to the hematopoietic aging and potential possibilities for delaying this process and possible rejuvenation.  A few concerns can be expressed.

  1. Lots of data and theories are based on the mouse studies, although some part is done with human samples. It would be helpful to make this disclaimer at the beginning and be clear through the text whether some particular pathways and mechanisms suggested for murine or human hematopoiesis (or both). It is also important to draw difference in aging process in human and mice in general, mice live much shorter lives and they are not ideal models for human aging. 
  2. Clonal hematopoiesis is mentioned at the beginning but did not play any role in the review, it could be further reduced. It is specifically human feature not reported for mouse. 
  3. Regarding rejuvenation authors could add a bit more information not only on attempts to improve  the HSCs properties but also on expanding pool of HSCs, which is a part of the anti aging therapy as well. Surely it needs to draw distinction between human and murine hematopoiesis, especially when some chemical compounds were tested.
  4. Typo at the line 192. Too many "that"

Reviewer 3 Report

Comments and Suggestions for Authors

Cui et al nicely report the mechanisms of HSC aging and perspective on their study and manipulation. The review is up to date, informative, well written and illustrated. I thanks the authors for this review.

Author Response

Thanks a lot for the nice comments.

Round 2

Reviewer 1 Report

Comments and Suggestions for Authors

Dear authors,

thank you for the careful revision of your manuscript. I still believe that 'adaptation' of the Figure does not mean straightforward copying. I would also prefer more precise using of the paper you're citing. Anyway, I won't hinder the publication further.